# Stage III Non-Small-Cell Lung Cancer: An Overview of Treatment Options

**Francesco Petrella** [1,2,*], **Stefania Rizzo** [3,4], **Ilaria Attili** [5], **Antonio Passaro** [5], **Thomas Zilli** [4,6,7], **Francesco Martucci** [6], **Luca Bonomo** [3], **Filippo Del Grande** [3,4], **Monica Casiraghi** [1,2], **Filippo De Marinis** [5] **and Lorenzo Spaggiari** [1,2]

1  Department of Thoracic Surgery, European Institute of Oncology IRCCS, 20141 Milan, Italy
2  Department of Oncology and Hemato-Oncology, University of Milan, 20141 Milan, Italy
3  Service of Radiology, Imaging Institute of Southern Switzerland (IIMSI), EOC, Via Tesserete 46, 6900 Lugano, Switzerland
4  Faculty of Biomedical Sciences, University of Italian Switzerland, Via Buffi 13, 6900 Lugano, Switzerland
5  Division of Thoracic Oncology, European Institute of Oncology IRCCS, 20141 Milan, Italy
6  Radiation Oncology, Oncological Institute of Southern Switzerland, EOC, 6500 Bellinzona, Switzerland
7  Faculty of Medicine, University of Geneva, 1211 Geneva, Switzerland
*  Correspondence: francesco.petrella@ieo.it; Tel.: +0039-0257489362

**Abstract:** Lung cancer is the second-most commonly diagnosed cancer and the leading cause of cancer death worldwide. The most common histological type is non-small-cell lung cancer, accounting for 85% of all lung cancer cases. About one out of three new cases of non-small-cell lung cancer are diagnosed at a locally advanced stage—mainly stage III—consisting of a widely heterogeneous group of patients presenting significant differences in terms of tumor volume, local diffusion, and lymph nodal involvement. Stage III NSCLC therapy is based on the pivotal role of multimodal treatment, including surgery, radiotherapy, and a wide-ranging option of systemic treatments. Radical surgery is indicated in the case of hilar lymphnodal involvement or single station mediastinal ipsilateral involvement, possibly after neoadjuvant chemotherapy; the best appropriate treatment for multistation mediastinal lymph node involvement still represents a matter of debate. Although the main scope of treatments in this setting is potentially curative, the overall survival rates are still poor, ranging from 36% to 26% and 13% in stages IIIA, IIIB, and IIIC, respectively. The aim of this article is to provide an up-to-date, comprehensive overview of the state-of-the-art treatments for stage III non-small-cell lung cancer.

**Keywords:** lung cancer; stage III; surgery; medical treatment; radiotherapy

## 1. Introduction

Lung cancer is the second most commonly diagnosed malignant tumor and the first cause of neoplastic death worldwide [1]. The most common histological type is non-small-cell lung cancers (NSCLC), accounting for 85% of all lung cancer cases. About 30% of new NSCLC cases are diagnosed at a locally advanced stage, which encompass a wide group of different clinical scenarios with a heterogeneous spectrum of therapeutic options [2]. In particular, stage III NSCLC is a highly heterogeneous group of lung tumors with significant differences in tumor size, local infiltration, and lymph nodal involvement. To be more specific, as shown in Table 1, stage III A NSCLC includes T3N1, T4N0, and T4 N1 diseases; stage IIIB NSCLC includes T3N2 and T4N2 diseases; and stage III C NSCLC includes T3N3 and T4N3 diseases [3].

Stage III NSCLC therapy is based on the pivotal role of multimodal treatments, including surgery, a wide-ranging option of systemic treatments, and radiotherapy; although the main scope of treatments—in this setting—is potentially curative, overall survival rates are still poor, ranging from 36% to 26% and 13% in stages IIIA, IIIB, and IIIC, respectively [3–6].

The ideal management of stage III NSCLC requires proper clinical and pathological staging, in particular of hilar and mediastinal lymph nodes, to correctly candidate patients to surgical approach or combined chemo–radiotherapy [7,8]. Imaging studies including ultrasound, computed tomography (CT) scan and 18FDG positron emission tomography (PET) scan are widely used to stage and follow-up many different tumors. In the specific setting of lung cancer staging, mediastinal staging by endobronchial ultrasound (EBUS) transbronchial needle aspiration (TBNA) has been introduced and has nowadays almost completely replaced mediastinoscopy. All the above mentioned techniques represent the cornerstone of a proper approach to locally advanced NSCLC [8–16].

**Table 1.** Descriptors of tumors (T) and nodes in TNM eighth edition, to define stage III lung cancer (adapted from [3]).

| T | N0 | N1 | N2 | N3 |
|---|---|---|---|---|
| T1a | | | IIIA | IIIB |
| T1b | | | IIIA | IIIB |
| T1c | | | IIIA | IIIB |
| T2a | | | IIIA | IIIB |
| T2b | | | IIIA | IIIB |
| T3 | | IIIA | IIIB | IIIC |
| T4 | IIIA | IIIA | IIIB | IIIC |

## 2. Surgery

Therapeutic options depend on patient performance status, cardiopulmonary function, tumor characteristics and disease extent. The standard approach—for the vast majority of these patients—is chemoradiotherapy (CRT), although for some patients—with low volume stage III disease—surgery may represent an additional therapeutic option, in particular within a multimodality approach [17–20]. Surgical approach can be considered—within a multimodality approach—in stage IIIA and in very selected cases of stage IIIB; on the contrary it is not recommended in stage III C (N3 disease).

### 2.1. Incidental III A Disease (Intraoperative)

Patients with NSCLC—classified as stage I or II disease at clinical and pathological staging might be found to present an incidental, unforeseen intraoperative N2 involvement. In this case, postoperative chemotherapy is strictly indicated and these patients present a relatively good prognosis [21]. In this setting, adjuvant chemotherapy after radical resection reduces possible relapse due to micro metastases. On the other hand, the role of postoperative radiotherapy is still debated: in fact, it is not clear whether adjuvant radiotherapy may improve the outcome of such patients. In any case, if a case–by–case evaluation of loco-regional risks discloses a higher chance of local relapse, adjuvant radiotherapy can be taken into consideration as a valuable additional therapeutic option [22]. It can be offered either after postoperative chemotherapy or as concurrent adjuvant chemoradiotherapy [23]. In case of incomplete resection—both with microscopic (R1) or macroscopic (R2) residual disease—adjuvant thoracic radiotherapy or concurrent chemoradiotherapy should be carefully discussed individually by the multidisciplinary team, as no clear guideline currently exists since the number of these patients is extremely small [22].

### 2.2. Potentially Resectable IIIA(N2) Disease (Preoperative)

Several multimodality therapeutic approaches have been proposed in patients with preoperatively diagnosed IIIA(N2) disease including preoperative chemotherapy followed by surgical resection [23–30], preoperative chemoradiotherapy followed by surgical resection or definitive chemoradiotherapy without surgery [20,31–40].

Only definitive concurrent chemoradiotherapy protocol versus preoperative concurrent chemotherapy and radiotherapy followed by surgical resection has been studied in a prospective randomised trial, comparing potentially resectable stage IIIA (N2) patients [41].

No difference in overall survival (OS) by intent to treat analysis was observed between the two groups; on the other hand, a better progression free survival (PFS) for patients receiving with surgical resection was reported [41]. Although both strategies remain possible treatment options in this setting, on the basis of the final results of this clinical trial—it has been observed a 26% mortality rate in the right-sided pneumonectomies, which is much higher than expected for this procedure [22]. On the other hand, recent papers—focusing on perioperative mortality after induction treatments and pneumonectomy-disclosed a much more acceptable overall 30-day mortality of 7% [42–47].

Another trial by the Swiss Group (SAKK) enrolled cytologically or histologically proven IIIA (N2) patients which were then randomized into preoperative chemotherapy followed by surgery versus sequential neo-adjuvant chemotherapy and then radiotherapy followed by surgery; the primary and the secondary endpoints of the study were OS and PFS, respectively; no significant differences were observed between the two arms [48,49].

In the light of these different trials results, it is generally accepted that—in this complex clinical scenario—a pivotal role in the decisional process is played by the multidisciplinary team, including an expert thoracic surgeon—for an upfront decision about the whole therapeutic pathway, thus avoiding any split in the radiotherapy application, when needed [22].

### 2.3. Potentially Resectable IIIA(N2) Disease and Selected IIIB Disease (at High Risk of Incomplete Resection)

For potentially resectable Pancoast tumor, concurrent preoperative chemo–radiotherapy followed by surgery represents the standard of care [48], although our personal experience suggests that surgical resection after induction chemotherapy offers valuable results [50–52].

A similar approach using concurrent chemo–radiotherapy to reduce the primary tumor and down-stage the disease can be applied to certain T3 N2 or T4 N0-1 tumors [20,33–35], although—even in this setting—our preference is surgical radical resection after induction chemotherapy [53–57]. A German trial (ESPATU) comparing surgery versus definitive chemoradiotherapy boost following complex induction chemotherapy and concurrent chemoradiotherapy did not disclose any benefit in OS or PFS for surgical resection, but both study arms showed excellent long-term survival results [58]. Radical resection in stage III NSCLC preferably includes parenchyma saving procedures such as sleeve resection, lobectomy, or bi-lobectomy avoiding—whenever possible—pneumonectomy [45]. On the other hand, it has recently become clear—and it is well accepted—that in very selected cases, radical resection will require a pneumonectomy or a tracheal sleeve pneumonectomy, which can be safely performed in experienced high-volume centers [42,43,59–61]. Post-operative mortality in stage III NSCLC receiving surgical treatment is reported to range between 2–3% for lobectomy and 3–8% for pneumonectomy [45]; as a clear relation between postoperative outcome and volume of surgical procedures has been reported, these procedures should be restricted to high-volume experienced centers [45].

### 3. Systemic Treatments

In the complex and varied scenarios that might occur within the stage III NSCLC definition, the use of systemic therapies virtually always represents a mainstay of treatment for patients with such a diagnosis [62].

Until recently, platinum-based chemotherapy was the only standard systemic treatment considered for stage III NSCLC, either resectable or not, with limited benefit both as neoadjuvant/adjuvant (absolute survival benefit about 5% at 5 years compared with placebo; in stage III: HR for neoadjuvant 0.84, 95% CI, 0.75–0.95 | HR for adjuvant 0.83, 95% CI, 0.72–0.94) [21] and as a definitive treatment in combination with concurrent or sequential radiotherapy (5-year survival rate of 10–20%) [31,63]. However, recent evolving improvements in biomarker evaluation and patients' selection rapidly moved the investigation on the use of immunotherapy and targeted agents from the advanced disease also to early-stage and stage III, as part of integrated treatments, highlighting the importance of adequate staging and a multidisciplinary approach in this setting [64]. Indeed, the final

aim in the treatment of patients with locally advanced NSCLC may vary consistently from curative to palliative intent (e.g., for the control of pain or symptoms from mediastinal involvement) [62,65].

### 3.1. Potentially Resectable Stage III Disease

As previously discussed, when stage III NSCLC is considered potentially resectable according to TNM 8th, the goal of the multimodal treatment is curative. In particular, after discussion within the multidisciplinary team, patients should become candidates for induction systemic treatment [62,64]. In those minoritarian cases with unforeseen N2 involvement before primary surgery for T1–T3 tumors, adjuvant systemic therapy is indicated [62].

### 3.2. Immunotherapy in the Perioperative Setting

In the adjuvant setting, treatment with adjuvant atezolizumab for one year after platinum-based chemotherapy (1 to 4 cycles) has been demonstrated to improve disease-free survival (DFS) compared with best supportive care, in patients with resected stage II-IIIA (7th AJCC) whose tumors have positive programmed death ligand-1 expression (PD-L1 $\geq$ 1%) (median DFS not estimated for 35.3 months; hazard ratio [HR], 0.66; 95% CI, 0.50 to 0.88; $p$ = 0.004), and in particular with high PD-L1 ($\geq$50%) (median DFS not estimated for 35.7 months; HR, 0.43; 95% CI, 0.27 to 0.68) [66]. With an updated median follow-up of 46 months, an interim analysis demonstrated a trend toward improved overall survival among patients with PD-L1 $\geq$ 1%, stage II–IIIA NSCLC, with particular benefit observed in PD-L1 $\geq$ 50% (HR 0.43, 95% CI: 0.26–0.71), resulting in a 5-year survival rate of 84.8% versus 67.5% in this population [67]. Based on these results, adjuvant atezolizumab received approval by the Food and Drug Administration (FDA) in resected stage II-IIIA PD-L1 positive NSCLC, and by European Medicines Agency (EMA) in PD-L1 high, after standard adjuvant chemotherapy, in patients without EGFR mutation or ALK rearrangement. Other immune checkpoint inhibitors (ICIs) are under evaluation in the adjuvant setting, with less strong and controversial efficacy data to date [64,68]. Moving forward, strong preclinical rational and growing clinical evidence support the greater activity of ICIs when administered in the neoadjuvant rather than adjuvant setting. Indeed, the presence of an onsite tumor accounts for a greater potential for T-cell activation and antitumor activity [69,70]. With this background, early phase trials were conducted to evaluate anti-programmed cell death protein-1/PD-L1 and anti-cytotoxic T-cell lymphocyte-4, either alone or in combination in stage I-III NSCLC. The results of these trials showed that the ICI treatment had no adverse effects on surgical outcomes, with previously unreported response rates ranging from 14–45% and a pathological complete response (pCR) up to 16% with single agent anti-PD-1/PD-L1 [70,71]. The pCR rates were up to 29% when anti-PD-1/PD-L1 were used in combination with anti-CTLA-4, but with relevant safety signals [72]. Subsequently, different combinations of chemotherapy and ICI in a neoadjuvant setting were assessed. Some of the reasoning behind this is that chemotherapy might be able to destroy cancer cells and cause them to rupture, leading to the release of antigens that the immune system can recognize and target.

With particular concern for stage IIIA NSCLC, the phase 2 NADIM study looked at the combination of three cycles of nivolumab with carboplatin and paclitaxel, followed by surgery and then adjuvant nivolumab for 1 year [73]. Patients with stage IIIA non-small-cell lung cancer and without specific mutations were included, where 89% of the participants were successfully operated on, with none having a progression of the disease during treatment. Here, 83% saw a major pathological response (MPR) in the tissue from surgery and 63% had no cancer cells left in the tissue samples. Interestingly, CT scans showed a lower percentage of complete or partial responses compared with the tissue samples. In this study, 77.1% of patients were progression free after 24 months [73]. The randomized trial Check Mate 816 included 358 patients with stage Ib ($\geq$4 cm)-IIIA NSCLC with no mutations in EGFR and ALK. The study assigned the patients to have either immunotherapy plus

chemotherapy or chemotherapy alone before surgery. Immunotherapy plus chemotherapy resulted in more people being able to have the surgery (83%) and they also had a higher pCR (24% compared with 2% *p* < 0.0001) [74]. The experimental arm also had better event-free survival and overall survival, although the results were not quite statistically significant yet. Because of the findings of the study, the combination of chemotherapy and nivolumab was approved by the FDA in March 2022 as a neoadjuvant treatment for resectable NSCLC ≥ 4 cm, including stage III [74].

With respect to surgical implications, a meta-analysis of neoadjuvant ICI trials showed a resection rate of 85.8% with an average surgical delay of 3.8–7.4% and a conversion rate (from minimally invasive to open) of 17.4% [75]. In the CheckMate 816 trial, patients who received a combination of chemotherapy and immunotherapy were more likely to have minimally invasive surgery (77%) and less likely to have pneumonectomies (17%) than those who only had chemotherapy (61% and 25%, respectively) [74].

### 3.3. Targeted Agents in the Perioperative Setting

In non-squamous NSCLC, the identification of targetable driver gene alterations with a significant impact on survival in the advanced stage has led to evaluating the use of targeted treatments with small molecule tyrosine kinase inhibitors (TKIs) in the perioperative setting (neo or adjuvant or combined), with different trials ongoing enrolling patients with EGFR or ALK positive tumors, but also ROS1, NTRK, and BRAF V600E [76–78]. To date, a paradigmatic change in the adjuvant setting resulted from the ADAURA trial, where patients with resected stage IB-IIIA (7th AJCC) NSCLC harboring EGFR common alterations (del19 or p.L858R) were randomized to receive osimertinib, an oral third generation EGFR TKI, or placebo for 3 years, after platinum-based chemotherapy, administered according to the physician's choice [76]. The trial reported a significant DFS improvement with osimertinib in the intention-to-treat population, with HR 0.23 (95% CI 0.18–0.30) in stage II-IIIA, primary endpoint population, and significantly lower incidence of CNS recurrence: CNS-DFS HR 0.24 in stage II-IIIA (95% CI 0.14–0.42) [77]. These data led to marketing authorization for treatment in the adjuvant setting for patients with stage IB-IIIA NSCLC harboring EGFR exon 19del or exon 21 p.L858R mutation. With specific respect to stage IIIA, the four-year DFS rate with osimertinib was 65% versus 14% with placebo (HR 0.20, 95% CI 0.14–0.29) [79]. Such results were obtained in resected stage IIIA patients who did not receive previous neoadjuvant treatment, thus suggesting the possibility to reconsider the indication of neoadjuvant chemotherapy in those patients with technically resectable N2 disease who are known to harbor del19 or L858R EGFR mutation, in order not to preclude such a valid post-surgical treatment option.

### 3.4. Unresectable Stage III Disease

The management of unresectable stage III NSCLC usually involves a combination of chemotherapy and radiation, with concurrent schedules being the preferred method when available. Recently, it has been approved to use immunotherapy consolidation with durvalumab administered for 12 months as an additional treatment. Indeed, in the phase III PACIFIC trial, 709 patients were randomized to receive durvalumab or placebo, every two weeks, for a period of up to 12 months [80]. The five-year rates for OS (42.9% versus 33.4%) and PFS (3.1% versus 19.0%) were higher with durvalumab than with the placebo. The benefit of the drug was seen, regardless of PD-L1. Furthermore, no extra toxicity (side-effects) was seen when taking durvalumab. However, based on a post hoc analysis, drug approval in Europe was only limited to people with tumors who were positive for PD-L1 (≥1%) [81]. Several different approaches are being explored to further improve survival among patients with stage III NSCLC, and a comparison of the different results between trials is challenging because of how varied the delivery time for the ICIs can be, and also because of the different eligibility criteria for each trial. Additionally, the window for enrollment varied for each trial as well. As for resectable disease, targeted treatments with TKIs are being evaluated in unresectable stage III NSCLC with actionable genomic

alterations (e.g., osimertinib after definitive chemo-radiation in the LAURA trial) [82]. Future developments and the results of the different strategies that have been investigated will potentially further change the paradigm approach to stage III NSCLC [83].

## 4. Radiotherapy

Radiotherapy represents one of the mainstay treatments for patients with stage III NSCLC, although its role and timing differs depending on the extent and location of the disease [3,7].

For superior sulcus (Pancoast) tumors, Buderi et al., in a systematic review including 550 patients [84], showed that induction chemo–radiotherapy followed by surgery resulted in superior OS compared with radiotherapy followed by surgery or surgery alone (five-year OS ranging between 36.4 and 84%; 11 and 49%; and 20 and 30%, respectively). Two prospective trials confirmed this approach. In the Southwest Oncology Group Trial 9416 (Intergroup Trial 0160) [85], 111 patients were treated with concomitant neo-adjuvant chemotherapy (two cycles of cisplatin and etoposide) and radiotherapy (45 Gy in 25 fractions) followed by surgical resection and two additional cycles of adjuvant chemotherapy. The five-year OS rate was 44% for the whole population, increasing up to 54% after complete resection. The Japanese 9806 phase II prospective trial confirmed these results with a reported five-year DFS and OS rate of 45% and 56%, respectively [86].

For patients with potentially resectable N2-NSCLC disease, the use of neoadjuvant chemoradiotherapy was not associated with an OS or PFS benefit compared with induction chemotherapy alone, despite higher rates of mediastinal downstaging, complete pathologic response of mediastinal lymph nodes, and higher rates of R0 resections observed in patients receiving preoperative radiotherapy [18,87,88]. In the postoperative setting, two contemporary clinical trials failed to demonstrate a better outcome with postoperative radiotherapy (PORT) in patients with a completely resected NSCLC disease with mediastinal N2 involvement and who had received neoadjuvant or adjuvant platinum-based chemotherapy. In the Lung ART trial, 501 patients with completely resected NSCLC with pathologically proven N2 disease were randomly assigned to receive PORT (252 patients, 54 Gy in 27 to 30 fractions) or no PORT (249 patients). In this trial, 3-year DFS (47.1% vs. 43.8%) and OS (66.5% vs. 68.5%) rates were similar between patients treated with or without PORT, despite a lower rate of mediastinal relapse (46.1% vs. 25%). On the other hand, grade 3–4 pneumonitis (5% vs. <1%) and late cardiopulmonary toxicity (11% vs. 5%) were higher in patients treated with PORT compared with the control arm [89]. These findings were confirmed by the phase III PORT-C trial. No DFS or OS benefit was observed with PORT compared to the observation in 394 patients with pIIIA-N2 NSCLC who underwent complete resection and adjuvant platinum-based chemotherapy [90]. On the other hand, PORT may be considered in selected patients with positive surgical margins and extracapsular extension (both excluded in the Lung ART trial), based the potential OS benefit observed with PORT in this specific subpopulation [91].

As far as NSCLC patients with N2 unresectable or with N3 lymph nodes are concerned, definitive concurrent chemo–radiotherapy is the standard-of-care treatment, as demonstrated by the EORTC 08941 [22] and ESPATUE trials [58] and by a meta-analysis of six randomized trials [92]. A platin doublet with radiotherapy up to a total dose of 60 Gy should be offered to these patients [93,94], with consolidation therapy with durvalumab for up to 12 months in patients without disease progression, as per the PACIFIC trial [80,95]. Because of the high rates of local and regional recurrence after concurrent radio–chemotherapy, different strategies have been attempted in order to enhance the efficacy of radiotherapy treatments: increasing the total delivered dose [96–99], hypofractionation [100,101], boosting the dose in more active tumour areas [98,99], different dose levels depending on PET avidity [102,103], and hyperfractionation [104]. Despite these assumptions, the RTOG 0617 randomized phase III trial failed to demonstrate a benefit from dose escalation, showing worse outcome results with 74 Gy in 37 fractions of 2 Gy compared with the standard dose of 60 Gy in 30 fractions [105]. On the other hand, modern

radiotherapy techniques such as intensity-modulated radiation therapy (IMRT) decreased the incidence of severe pneumonitis compared with conformational three-dimensional techniques (3.5% vs. 7.9%) and limited the doses delivered to the heart [89].

In poor-risk patients who are not candidates for standard combined-modality therapy, radiotherapy alone results in a median OS of 10 months only, with an estimated OS rate at 5-year of 5% [106–109]. Nevertheless, in this population, exclusive radiotherapy at doses ranging between 40 and 50 Gy provide a modest, but significant, survival advantage at one year compared with observation (18 vs. 14%) [110].

The number of patients and outcome data with confidence intervals, where available, are summarized in Table 2.

**Table 2.** Number of patients and outcome data with confidence intervals, for the included articles, where available.

| Article | Number of Patients | Outcome Data | 95% Confidence Intervals |
|---|---|---|---|
| Goldstraw [3] | 8275 | ND | ND |
| Kim [45] | 2126 | 30-day mortality (right vs. left pneumonectomy) = 1.97 OR | 1.11–3.49 |
| Pignon [21] | 4584 (628 stage III) | Absolute effect of chemotherapy on OS (5.4% at 5 years). For stage III HR 0.83 | 0.72–0.94 (for stage III) |
| Felip [66] | 1280 | The stratified HR for DFS was 0.66 | 0.50–0.88 |
| Wakelee [67] | 1005 | When PD-L1 $\geq$ 50% HR 0.43 | 0.26–0.71 |
| Forde [70] | 21 | The rate of recurrence-free survival at 18 months was 73% | 53–100 |
| Cascone [71] | 53 | The primary endpoint of MPR, was observed in 5/23 of patients in the nivolumab arm | 7–44 |
| Forde [74] | 358 | The median event-free survival was 31.6 months with nivolumab plus chemotherapy | Not reached |
| Jiang [75] | 988 | In patients treated by neo-adjuvant immunotherapy, resection rate of 85.8% with an average surgical delay 7.4% | NA |
| Tsuboi [76] | 224 | DFS improvement with osimertinib in the intention-to-treat population, HR 0.23 | 0.18–0.30 |
| Wu [77] | 682 | The 4-year DFS rate with osimertinib was 65% versus 14% with placebo; HR 0.20 | 0.14–0.29 |
| Faivre-Finn [80] | 709 | OS (HR = 0.71) PFS (HR = 0.55) | 0.57–0.88 0.44–0.67 |
| Buderi [84] | 550 | Induction chemo–radiotherapy followed by surgery, RT followed by surgery, and surgery alone resulted in 5-year OS ranging between 36.4–84%; 11–49%; 20–30%, respectively | NA |
| Rusch [85] | 110 | The 5-year OS rate was 44% for the whole population, 54% after complete resection | ND |
| Kunitoh [86] | 76 | The 5-year OS rate was 56% The 5-year PFS rate was 45% | ND |
| Le Pechoux [89] | 501 | The 3-year DFS was 47% with PORT versus 44% without PORT | 40–54 |

ND = not declared; NA = not applicable; OR = odds ratio; OS = overall survival; DFS = disease free survival; PFS = progression free survival; HR = hazard ratio; CI = confidence interval; MPR = major pathological regression.

## 5. Future Perspectives Offered by Imaging

In recent years, advances in imaging options have been applied to many oncologic imaging settings, including lung cancer [10,14,111–113]. Interesting new evidence and hints have indeed studied the possibility for the early identification of patients who may benefit from target therapies and immunotherapy, as well as to predict what patients will respond to therapies or will develop complications, through the use of artificial intelligence aided techniques. Artificial intelligence (AI) is an umbrella term encompassing any technique that enables computers to mimic human intelligence. Machine learning is part of artificial intelligence, and describes algorithms that self-improve after further exposure to data. Deep learning is a specific subtype of machine learning, mainly relying on convolutional neural network techniques [114]. Radiomics is an emerging translational field of research aiming to extract high-dimensional data from clinical images [115], that can rely on machine learning and deep learning techniques to help with the construction of meaningful predictive models. For example, Sun et al. aimed to develop and validate a radiomics-based biomarker of tumor-infiltrating CD8 cells in four cohorts of patients, including lung cancer patients, enrolled into clinical trials to undergo anti-programmed cell death protein (PD)-1 or anti-programmed cell death ligand 1 (PD-L1) monotherapy [116]. The authors demonstrated that in patients treated with anti-PD-1 and PD-L1, a high baseline radiomic score was associated with a higher proportion of patients who achieved an objective response at 3 months and a higher proportion of patients who had an objective response or stable disease at 6 months. A high baseline radiomic score was also associated with improved overall survival in univariate and multivariate analyses [116]. In another interesting study, Lou et al. aimed to identify radiation sensitivity parameters that can predict treatment failure and thus guide the individualization of radiotherapy dose [117]. To this end, they queried pre-therapy lung CT images into a multitask deep neural network that incorporates radiomics into the training process, and then combined these data with clinical variables. So, they derived an individualized radiation dose resulting in an estimation of failure probability below 5% at 24 months [117]. However, in their study, they included all stages of lung cancer, and it is not possible to derive specific data for lung cancer patients in stage III. Hosny et al. investigated the ability of deep learning networks to quantify radiographic lung cancer characteristics and to predict overall survival likelihood [118]. They designed a rigorous analytical setup with seven large and independent datasets of 1194 non-small-cell lung cancer patients staged I-IIIb, imaged by computed tomography across five institutions, to discover and validate the prognostic power of a convolutional neural network in patients treated with radiotherapy and surgery [118]. The authors demonstrated that in patients who were treated using surgery, deep learning networks significantly outperformed models based on predefined tumor features, as well as tumor volume and maximum diameter [119]. Furthermore, they demonstrated that the study of the areas around the tumor had the largest contributions to the prognostic signature [118]. In a specific group of 126 stage III lung cancer patients treated with radio–chemotherapy, Li et al. demonstrated that dual-omics features from different lung functional regions can improve the prediction of radiation pneumonitis [119]. Yoo et al. assessed a machine learning model that was able to predict pathological complete response after treatment with neoadjuvant chemoradiotherapy by analyzing the texture features from pre- and post-treatment PET-CT studies of patients with stage III non-small-cell lung cancer [120].

All of the abovementioned findings encourage further prospective studies validating their utility in lung cancer patient stratification and the development of personalized cancer treatment plans. In fact, while medical imaging has always provided an individual assessment of ailments, AI algorithms based on imaging biomarkers promise accurate patient stratifications and enable new research avenues for personalized healthcare. Although the use of AI to better understand and treat lung cancer patients has shown interesting results, some of which have been mentioned above, the implementation of these algorithms in clinical practice still faces obstacles, including mainly technical difficulties, the need for validation, and regulatory aspects. For instance, patient records exist in multiple forms, such

as free text, recorded speech, or medical images, and are rarely appropriately organized for computational analysis. The possibility to access large amounts of well-organized high-quality data is essential for the application of AI techniques in health care. Furthermore, stakeholders may not be willing to share data due to responsibilities related to personal privacy laws [121,122].

## 6. Final Considerations

The vast majority of clinical studies have only enrolled patients presenting performance status (PS) scores ranging from 0 to 1, with few patients with PS scores of 2 or higher; usually, patients presenting with a PS score of 3 or higher are excluded. For this reason, supportive care alone is usually recommended for patients with an advanced PS score. On the other hand, in daily clinical practice, approximately 25% of lung cancer patients present with a PS score of 3 or 4 at the beginning of treatment or attain scores between 3 and 4 during the course of therapy. Many patients with higher PS scores can be successfully treated using individualized anti-tumor treatment and additional life-support strategies. For example, patients diagnosed with early-stage NSCLC and suffering from other underlying cardiopulmonary comorbidities can be offered many advanced support technologies that may optimize preoperative assessment, intraoperative protection, and postoperative support.

Moreover, radiotherapy can be tailored in patients with concurrent severe comorbidities to pursue a radical approach in patients who cannot tolerate surgical resection in the early stages diseases, can be effectively combined with systemic treatment in locally advanced stages, and can be offered as a tailored palliative option in advanced stages to improve symptom management [123].

With regard to the risk factor assessment profile, the available findings for stage III NSCLC suggest that, on the whole, the distribution of risk factors at this stage is analogous to that for lung cancer (there is a high percentage of smokers (58%), while the percentage of never smokers is quite low, ranging from 4% to 11.1%) [124].

With respect to residential radon, there are no published studies on its distribution by stage at diagnosis [125].

## 7. Conclusions

Stage III lung cancer includes a heterogeneous group of patients with significant differences in terms of tumor volume, local diffusion, and lymphnodal involvement. Although the survival rate is still poor (13–36% according to the different subgroups), stage III NSCLC can be treated with a curative intent. For this reason, in these patients, a pivotal role relies on multimodal treatment, including surgery, radiotherapy, and multiple options of systemic treatments. Furthermore, surgeons, oncologists, radiotherapists, and radiologists may be willing to integrate machine learning tools into the clinical care of lung cancer patients, joining a digital revolution that will help to provide a consistent, timely, and personalized treatment strategy to eventually improve patient outcomes.

**Author Contributions:** Conceptualization, F.P., S.R., M.C. and I.A.; methodology, F.P., S.R., I.A., T.Z. and F.M.; software, F.P. and S.R.; validation, all authors.; formal analysis, not applicable; investigation, F.P., S.R., I.A., T.Z., F.M. and L.B.; resources, S.R.; data curation, F.P., S.R., I.A., A.P., M.C., T.Z. and F.M.; writing—original draft preparation, F.P., S.R., I.A. and T.Z.; writing—review and editing, all authors; visualization, F.P.; supervision, F.D.G., F.D.M. and L.S.; project administration, S.R.; funding acquisition, not applicable. All authors have read and agreed to the published version of the manuscript.

**Funding:** This research received no external funding.

**Conflicts of Interest:** The authors declare no conflict of interest.

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
