# Peer review of "Stage III Non-Small-Cell Lung Cancer: An Overview of Treatment Options"

_curroncol, doi:10.3390/curroncol30030239_

Round 1
Reviewer 1 Report
This manuscript is organized in a schematic, exhaustive, complete structure, and seems to discuss each different treatment options very well.
I think that the authors have to underline how the optimal treatment management must be tailored to the characteristics of the patient (e.g. unfit patients for surgery, pulmonary toxicity from radiotherapy in operated patients).
Reviewer 2 Report
I had the great pleasure to review this manuscript submitted by Dr Petrella et al.
The manuscript is very well written and discuss multiple interesting point in such a very challenging topic. The authors has covered in excellent way all aspects of stage III lung cancer treatment options.
I enjoyed reading the paper and found extremely interesting.
Reviewer 3 Report
The purpose of this article is an overview of the latest research in lung cancer and recommendations by a large group of experts, stating that “the aim of this article is to provide an up-to-date, comprehensive overview of the state of the art treatments for stage III non-small cell lung cancer,”
This is not structured as a systematic review, and it would be unfair to request that depth of critical analysis of the trials.
[Nevertheless, if I were a physician, surgeon or patient I would welcome basic Cochrane Risk of bias data about the trials including the number of eligibles, patients randomised, completeness of delivery of the interventions, and an intention-to treat analysis.
I listed below all the instances in which outcome data were presented. Would it be possible please to generate a table with numbers of patients and outcome data with 95% confidence limits?]
.”overall survival rates are still poor, ranging from 36% to 26 % and 13% in 52 stages IIIA, IIIB, and IIIC, respectively [3 – 6].”
“it has been observed a 26% mortality rate in the right-sided pneumonecto- 97 mies, which is much higher than expected for this procedure [22].”
”Post-operative mortality in stage III NSCLC receiving surgical 130 treatment is reported to range between 2%–3% for lobectomy and 3%–8% for pneumo- 131 nectomy [61];”
“Until recently, platinum-based chemotherapy was the only standard systemic treatment 139 to consider for stage III NSCLC, either resectable or not, with limited benefit both as neo- 140 adjuvant/adjuvant (absolute survival benefit about 5% at 5 years compared to placebo; 141 in stage III: HR for neoadjuvant 0.84, 95% CI, 0.75–0.95 | HR for adjuvant 0.83, 95% CI, 142 0.72-0.94) [21] and as definitive treatment in combination with concurrent or sequential 143 radiotherapy (5-year survival rate 10-20%)”
“in patients with resected stage II-IIIA (7th 161 AJCC) whose tumors have positive programmed death ligand-1 expression (PD-L1 ≥ 162 1%) (median DFS not estimated v 35.3 months; hazard ratio [HR], 0.66; 95% CI, 0.50 to 163 0.88; P = .004), and in particular with high PD-L1 (≥ 50%) (median DFS not estimated v 164 35.7 months; HR, 0.43; 95% CI, 0.27 to 0.68) [71]”
“improved overall survival 166 among patients with PD-L1≥ 1%, stage II–IIIA NSCLC, with particular benefit observed 167 in PD-L1≥ 50% (HR 0.43, 95% CI: 0.26-0.71), resulting in a 5-year survival rate of 84.8% 168 versus 67.5% in this population [72].”
“previously unreported 181 response rates ranging from 14-45% and pathological complete response (pCR) up to 182 16% with single agent anti-PD-1/PD-L1 [75-76]. pCR rate were up to 29% when anti-PD- 183 1/PD-L1 were used in combination with anti-CTLA-4, but with relevant safety signals”
“Immunotherapy 200 plus chemotherapy resulted in more people being able to have the surgery (83%) and 201 they also had a higher pCR (24% compared to 2% P < .0001) [79]. The experimental arm 202 also had better event-free survival and overall survival, although the results were not 203 quite statistically significant yet”
’a meta-analysis of neoadjuvant ICI trials showed a 207 resection rate of 85.8% with an average surgical delay of 3.8-7.4% and a conversion rate 208 (from minimally invasive to open) of 17.4% [80]. In the CheckMate 816 trial, patients 209 who received a combination of chemotherapy and immunotherapy were more likely to 210 have minimally invasive surgery (77%) and less likely to have pneumonectomies (17%) 211 than those who only had chemotherapy (61% and 25% respectively) [79].”
“Trial reported a significant DFS improvement with 223 osimertinib in the intention-to-treat population, with HR 0.23 (95% CI 0.18 - 0.30) in 224 stage II-IIIA, primary endpoint population, and significantly lower incidence of CNS 225 recurrence: CNS-DFS HR 0.24 in stage II-IIIA (95%CI 0.14-0.42) [81]. These data led to 226 marketing authorization for treatment in the adjuvant setting for patients with stage IB- 227 IIIA NSCLC harboring EGFR exon 19del or exon 21 p.L858R mutation. With specific re- 228 spect to stage IIIA, the 4-year DFS rate with osimertinib was 65% versus 14% with pla- 229 cebo (HR 0.20, 95% CI 0.14-0.29) [84]”
“The 5-year rates for OS 241 (42.9% versus 33.4%) and PFS (3.1% versus 19.0%) were higher with durvalumab than 242 with placebo.”
“superior sulcus (Pancoast) tumours, Buderi et al. in a systematic review including 550 263 patients [89] showed that induction chemo-radiotherapy followed by surgery resulted in 264 superior OS compared with radiotherapy followed by surgery or surgery alone (5-year 265 OS ranging between 36.4 and 84%; 11 and 49%; and 20 and 30%, respectively). Two pro- 266 spective trials confirmed this approach. In the Southwest Oncology Group Trial 9416 (In- 267 tergroup Trial 0160) [90], 111 patients were treated with concomitant neo-adjuvant chem- 268 otherapy (two cycles of cisplatin and etoposide) and radiotherapy (45 Gy in 25 fractions) 269 followed by surgical resection and two additional cycles of adjuvant chemotherapy. The 270 5-year OS rate was 44% for the whole population, increasing up to 54% after complete 271 resection. The Japanese 9806 phase II prospective trial confirmed these results with a re- 272 ported 5-year DFS and OS rate of 45% and 56%, respectively [91].”
“Lung ART trial, 501 patients with completely resected NSCLC with 282 pathologically proven N2 disease were randomly assigned to receive PORT (252 patients, 283 54 Gy in 27 to 30 fractions) or no PORT (249 patients). In this trial, 3-year DFS (47.1% vs 284 43.8%) and OS (66.5% vs 68.5%) rates were similar between patients treated with or with- 285 out PORT, despite a lower rate of mediastinal relapses (46.1% vs 25%). On the other hand, 286 grade 3-4 pneumonitis (5% vs…”
Reviewer 4 Report
Comments
1. In Introduction section, line 46, using of both words twice in same sentence is not advisable so please replace with the best alternate word.
2. The authors need to provide the flow chart/sketch of the stage III NSCLS categories and comparison. The authors need to create the flow chart/diagram of Non-invasive and Invasive mode of the therapeutic management by including the surgical approach.
3. The authors need to describe the combination of the alternate mode and novel treatment including chemotherapy/radiotherapy and targeted immunotherapeutic approach.
4. The authors need to describe the risk factor assessment profile and scoring dynamics with the survivals of the Stage III non-small cell lung cancer population
Reviewer 5 Report
Few References, in the text, are missing
(The Authors must see my remarks)

Round 2
Reviewer 3 Report
Thanks to the authors for the new table with study data and outcomes. This is very helpful in assesing the evidence in one place.
Reviewer 4 Report
The authors have made substantial updates in the revised manuscript by addressing the concerns and comments raised in the original manuscript. The revised manuscript may be considered for publication subject to addressing the overall required concerns and incorporating the new editings.